# Effectiveness of Therapeutical Interventions on the Scapulothoracic Complex in the Management of Patients with Subacromial Impingement and Frozen Shoulder: A Systematic Review

**DOI:** 10.3390/jfmk8020038

**Published:** 2023-03-27

**Authors:** Rosario Ferlito, Gianluca Testa, Kathryn Louise McCracken, Salvatore Moscato, Giovanni Maria Zerbito, Flora Maria Chiara Panvini, Chiara Blatti, Vito Pavone, Marco Sapienza

**Affiliations:** 1Department of Medicine and Health Science “Vincenzo Tiberio”, University of Molise C/da Tappino c/o Cardarelli Hospital, 86100 Campobasso, Italy; 2Department of Biomedical and Biotechnological Sciences, University of Catania, 95123 Catania, Italy; 3Department of General Surgery and Medical Surgical Specialties, Section of Orthopaedics and Traumatology, P.O. “Policlinico Gaspare Rodolico”, University of Catania, 95124 Catania, Italy; 4School of Medicine, University College Cork, College Road, T12 K8AF Cork, Ireland

**Keywords:** shoulder pain, subacromial impingement syndrome, frozen shoulder, scapular kinematics, manual therapy, scapular function

## Abstract

Shoulder pain is one the most common musculoskeletal complaints. The most common pathological causes of shoulder pain in the general population are subacromial impingement syndrome and adhesive capsulitis, commonly referred to as “frozen shoulder”. The purpose of this study was to evaluate the role of the scapulo-thoracic complex, particularly in scapular kinematic functions, in rehabilitative interventions for shoulder pain in patients suffering from these two common conditions. This systematic review was performed using the scientific search engines PubMed, PEDro and Cochrane Library, considering only randomized controlled clinical trials. Selected articles were evaluated according to the level of evidence and methodological quality. Thirteen randomized clinical trials were selected. Interventions have been divided into three macro-categories: (1) manual therapy in patients with subacromial impingement, (2) therapeutic exercise programs including interventions on the scapulothoracic complex in patients with subacromial impingement syndrome, and (3) therapeutic exercise programs including interventions on the scapulothoracic complex in patients with frozen shoulder. Following this, a qualitative analysis was performed according to outcomes such as pain, shoulder function, and scapular kinematics. Physiotherapy exercise programs that included scapular motor control training and scapular mobilizations, in particular, those of the scapulo-thoracic complex in scapular kinematic function, represent valid alternatives in the management of patients with subacromial impingement syndrome.

## 1. Introduction

Shoulder pain is a very common condition among the general population, with an estimated prevalence from 16% to 48%, and is the third most common musculoskeletal problem, following low back pain and neck pain [1,2]. The musculoskeletal issues most frequently associated with shoulder pain are Subacromial Impingement Syndrome (SIS) and frozen shoulder (adhesive capsulitis), which both share common pathomechanical components [3]. Unfortunately, for most patients, the symptoms can be persistent and very disabling, drastically limiting the function of the entire upper extremity, preventing the individual from performing normal activities of daily living both at work and at home [3]. Additionally, there is significant financial burden associated with these conditions. As the pain and disability progress, patients require more frequent and costly health care services. Simultaneously, their ability to work decreases, resulting in work absences or even retirement; thus, further worsening their financial position [3].

SIS is one of the most common shoulder diseases [4]. It seems to be more common in athletes or workers who repeatedly abduct their shoulders, but it can also occur in relatively sedentary individuals [5]. The pathomechanics of this condition are related to repeated and potentially damaging compression of the tissues contained within the subacromial space [6].

The tissues that are most affected by compression between the coraco-acromial arch and the head of the humerus are the tendon of the supraspinatus muscle, the tendon of the long head of the biceps brachii muscle, the upper portion of the capsule, and the subacromial bursa [6].

In a healthy, pain-free shoulder, full shoulder abduction occurs in conjunction with significant upward scapulothoracic rotation, which is usually associated with small scapular adaptation movements such as posterior tilt and external rotation [7]. Some studies have reported that that subjects with SIS have a lower than normal capacity for upward scapulothoracic rotation, less posterior tilt, and less external rotation of the scapula during shoulder abduction [7]. It is therefore believed that these abnormal kinematic aspects contribute to subacromial impingement as they reduce the free space between the head of the humerus and the coracoacromial arch [4].

Another factor that contributes to a reduction of the volume of the subacromial space is improper positioning of the scapula in relation to the thorax due to poor posture in the cervical and thoracic sections of the spine [8,9].

Similarly, many individuals with frozen shoulder also have a decrease in posterior tilt and upward rotation of the scapula during limb elevation. Further, this limitation is associated with exacerbated symptoms and a decrease in treatment success [10]. Thus, interventions focused on the scapula can be rationally justified in patients with frozen shoulder, given the common alterations in scapular kinematics previously mentioned [10].

In recent decades, the scientific literature has been trying to highlight the importance of physiotherapy interventions targeting the scapulo-thoracic complex. Therefore, the purpose of this study is to evaluate the role of the scapulo-thoracic complex, and particularly, in scapular kinematic function in rehabilitative interventions for shoulder pain in patients with subacromial impingement syndrome and frozen shoulder.

## 2. Materials and Methods

The research was conducted based on the PRISMA Statement guidelines, a protocol checklist designed to facilitate reporting in systematic reviews [11,12].

The systematic review was carried out using the search engines PubMed, PEDro, and the Cochrane library, considering clinical trials published in the last 10 years (2011–2021) to ensure an adequate representation of the most recent evidence available.

Only randomized clinical trials were included because their high quality of evidence strengthens the qualitative review.

The search was performed by combining the keywords in the search string with the Boolean operators “AND” and “OR”.

The search strings were as follows:PubMed: (Shoulder pain) AND ((Scapular kinematic) AND ((shoulder impingement) OR (Frozen Shoulder)) AND ((manual therapy) OR (feedback) OR (exercise therapy));PEDro: Title and Abstract: (shoulder pain) AND (scapular function); Problem: pain; Part of the body: upper arm, shoulder or shoulder girdle;
Subdiscipline: *musculoskeletal*;Method: *clinical trial*;Published since: *2011.*Cochrane library: *(scapular function) AND (shoulder pain) AND ((shoulder impingement) OR (frozen shoulder))*.

### 2.1. Studies Selection

The aforementioned strings yielded a total of 109 articles including: 10 articles from PubMed, 24 articles from PEDro, and 75 articles from Cochrane library (Figure 1). Firstly, 25 duplicate articles common among the searches were excluded. Next, articles were selected by title, and 49 articles that were not relevant to the search question were excluded. The abstracts of the remaining 35 articles were reviewed, and 11 articles were excluded for the reasons outlined in Figure 1. The remaining 24 full texts were searched: 11 were excluded because the full text was not available, while the remaining 13 full texts that met inclusion criteria were all included within the systematic review (Table 1).

### 2.2. Studies Included

Thirteen full texts [1,10,13,14,15,16,17,18,19,20,21,22,23] belonging to thirteen RCTs met the criteria of eligibility and were included in the review (Table 1).

### 2.3. Study Design

The included studies were all RCTs published in English over a period of approximately 8 years. The most recent one is from 2020 and the oldest one is from 2012 (Table 1). Search criteria included the years 2011–2021.

### 2.4. Type of Participants

Men and women older than 18 years with SIS who were part of the general population and not part of a specific subgroup of athletes, such as swimmers and volleyball players, were included. Men and women older than 18 years of age with frozen shoulder who have not had shoulder surgery were included. Those who have had shoulder surgeries were excluded from the study.

### 2.5. Type of Interventions

All studies that analyzed interventions aimed at the scapulo-thoracic complex, performed individually or supported by other physiotherapeutic interventions, were included, and those employing other interventions or no interventions were not included. Particular emphasis was given to articles that included the use of manual therapy interventions, as there is still much debate about their effectiveness.

### 2.6. Type of Outcomes

All studies that presented pain and function measured by questionnaires as outcomes were considered to be eligible. Additionally, studies that used computerized or other measures to analyze scapular kinematics were also included.

## 3. Results

Thirteen randomized clinical trials were selected for review. The interventions were divided into three macro-categories: (1) manual therapy in patients with subacromial impingement, (2) therapeutic exercise programs including interventions on the scapulothoracic complex in patients with subacromial impingement syndrome, and (3) therapeutic exercise programs including interventions on the scapulothoracic complex in patients with frozen shoulder. Manual therapy interventions in patients with subacromial impingement were used independently or in conjunction with classical therapeutic exercises. (1) Patients treated with thoracic manipulation, upward rotation and scapular tilt, or upward rotation and arm elevation demonstrated slightly better outcomes compared to the outcomes of those who were not treated. Otherwise there were no statistically significant differences related to the patients’ positioning (Table 2 and Table 3).

The use of thoracic thrusts for treating shoulder pain in patients with impingement did not have significantly better outcomes compared to those of the control group (Table 2).

(2) Improvements in pain, function, and scapular kinematics following various therapeutic exercise programs were evaluated. The addition of tactile and verbal feedback to a scapula-focused exercise program was shown to be very effective in improving pain and function compared to the levels of the control group, which performed the same exercises without feedback (Table 2). A slight improvement was observed in the humeral stabilization exercise treatment group compared to those of the control group. Greater external rotation and scapular posterior tilt were observed in the group who performed scapular stabilizations compared with those of the control group, which continued to be the case throughout the follow-ups. (3) Frozen shoulder patients were also evaluated for improvements achieved in pain, function, and scapular kinematics following various programs with therapeutic exercises. The different types of interventions were shown to be effective at improving pain. Although all the different interventions were shown to be effective for improving pain, the improvement was significantly larger in the group treated with scapular mobilizations compared to that of the control group. Similarly, patients treated with scapular PNFs and physical therapy and those treated with physical therapy demonstrated significantly greater improvements compared with those of the control group treated with classical exercises and physical therapy. Furthermore, scapular mobilizations were shown to provide greater shoulder function improvement.

**Table 2 jfmk-08-00038-t002:** Values of pain scales in the studies considered.

Study	Outcome Measure Tool	Results					
Study [1]	NPRS			Pre-intervention		Post-intervention	
		Thoracic thrust (average)		3.5		2.6	
		Simulated thrust (average)		3.6		2.4	
Study [17]	Penn Shoulder Score	Median of changes					
		Supine Thrust		Seated Thrust		Simulated Thrust	
		5.0 (2.0–7.0)		5.0 (1.5–8.2)		5.0 (2.0–9.8)	
Study [19]	NPRS			Pre-intervention		Post-intervention	
		Thoracic thrust (average)		3.3		2.4	
		Simulated Thrust (average)		2.4		2.2	
Study [20]	NPRS			Pre-intervention	2 days pre-intervention	2 days post-intervention	Follow-up
		Thoracic thrust (average)		3.3	2.5	2.4	2.4
		Simulated Thrust (average)		2.7	2.4	2.4	2.9
Study [22]	VAS at rest (from 0 to 100 mm)			Pre-intervention		Post-intervention	
		Exercise plus manual therapy (average)		19.3 ± 27.6		6.3 ± 11.6	
		Only exercise (average)		10.3 ± 14.1		3.6 ± 6.1	
Study [13]	VAS (on movement)		0 week	2 week	3 week	7 week	11 week
		Scapular mobilization group (average)	7.0	5.0	3.8	3.0	2.3
		False scapular mobilizations group (average)	7.3	5.6	4.8	3.8	3.7
		Group with exercises under supervision (average)	7.1	5.1	4.0	3.6	2.9
Study [14]	VAS (during activity)		Pre-intervention		6 week	12 week	
		Intervention group (average)	4.84		1.52	0.38	
		Control group (average)	5.32		2.36	1.26	
Study [15]	VAS (on movement)		Before treatment			After treatment	
		Intervention group (average)	5.7 ± 2.6			3.0 ± 1.9	
		Control group (average)	6.3 ± 1.9			5.1 ± 2.0	
Study [16]	NPRS		Pre-intervention		4 week	8 week	16 week
		Intervention group (average)	3.9		1.5	1.3	1.2
		Control group (average)	3.7		2.7	1.8	0.5
Study [18]	VAS		Before treatment			Follow-up	
		Treatment with feedback (average)	6.4			2.4	
		Treatment without feedback (average)	6.1			3.1	
		Control group (average)	5.7			6.2	
Study [21]	VAS		Before intervention			After intervention	
		Scapular PNF (average)	6.07			4.16	
		Classic exercises (average)	4.67			3.97	
		Control group (average)	6.55			4.22	
Study [23]	NPRS		Before intervention			After intervention	
		Intervention group (average)	8.00			3.93	
		Control group (average)	6.67			4.80	

### 3.1. Sample

The total number of participants with SIS enrolled was 522, of which 467 completed all the follow-ups.

The total number of frozen shoulder participants enrolled was 117, of which 115 completed all the follow-ups.

Forty-seven asymptomatic subjects also participated in the study [19], all of whom completed the treatment; this sample was not used in the review because it did not meet the inclusion criteria.

Among all studies, the minimum sample size was 22 participants [15], while the maximum sample size was 97 participants [19].

#### 3.1.1. Gender

There was a total of 243 men with impingement and 279 women with an M:F ratio of 0.871.

Forty-four men and seventy-three women with frozen shoulder were included, giving an M:F ratio of 0.603.

The asymptomatic group in the study [19] was comprised of 20 men and 27 women (Table 1).

#### 3.1.2. Enrollment

The studies included patients recruited through notices, advertising flyers, emails, in physiotherapy clinics, orthopedic practices, universities, orthopedic clinics, or through private orthopedic surgeons and physiotherapists.

#### 3.1.3. Setting

The studies took place in physical therapy departments, university clinics, and outpatient physical therapy clinics.

#### 3.1.4. Age

See Table 1 for the average age of the participants in various studies.

#### 3.1.5. Diagnostic Criteria

All the patients included in the SIS studies were diagnosed clinically. The probability of diagnosis was increased with multiple positive clinical signs in various combinations.

The clinical tests used were: Neer’s test, Jobe’s test, Hawkins–Kennedy test, pain on active elevation or abduction of the limb, pain during resistance to abduction or external rotation, and pain during palpation of the cuff tendons.

The participants in one study [14] also had the inclusion criteria of having scapular dyskinesia (at observation) and a positive *Scapular Assistance Test*.

The diagnosis of frozen shoulder was less consistent between studies; some relied on a clinical diagnosis, while others based the diagnosis on radiographic findings.

### 3.2. Type of Interventions

The interventions used by the studies were divided into three macro-categories, which grouped the approaches by type.

### 3.3. Approaches

Manual therapy in patients with SIS alone or performed in combination with stretching and strengthening exercises [22];Therapeutic exercise programs including interventions on the scapula-thoracic complex in patients with SIS [18];Therapeutic exercise programs including interventions on the scapula-thoracic complex in patients with frozen shoulder [21].

### 3.4. Contents of Therapeutic Interventions

#### 3.4.1. Manual Therapy in Patients with Subacromial Impingement Syndrome (SIS)

Manual therapy interventions in patients with impingement were administered independently or in conjunction with classical therapeutic exercises (Table 1 and Table 2).

In four studies [1,17,19,20], only thrust operations aimed at the thoracic spine were performed, while some focused on the lower segment [1], on the intermediate segment [1,19,20] and on the upper segment [8,9]. The thrusts were also performed in various positions: sitting [1,17,19,20], prone [1], and supine [17] (Table 1 and Table 2).

In study [22], three stretching and three strengthening exercises of the shoulder were combined with manual therapy interventions that included various techniques: III and IV degree mobilizations with arthrokinematic and osteokinematic movements for the glenohumeral, scapulo-thoracic, acromion-clavicular and sterno-clavear joints, and the cervical spine; then soft tissue techniques (deep frictions and kneading), PNF, rhythmic stabilizations and contraction, and relaxation techniques addressing the affected muscles (Table 1 and Table 2).

#### 3.4.2. Therapeutic Exercise Programs including Interventions on Scapulo-Thoracic Complex in Patients with SIS

These interventions were performed in combination with each other or with other types of therapist exercises aimed at the shoulder.

The following were performed:Exercises including isometry, isotonic, concentric and eccentric, and stretching, which were all focused on the scapula with tactile and verbal feedback [18];Manual mobilization, stretching, and training in motor control of the scapula [15];Strengthening exercises for the peri-scapular muscles combined with scapular stabilization exercises to emphasize scapular retraction and depression [16];A supervised combination of closed and open kinetic chain scapular stabilization exercises, with shoulder strengthening and stretching [14];Scapular mobilization, preceded by hot packs and TENS [13].

#### 3.4.3. Therapeutic Exercise Programs including Interventions on the Scapulo-Thoracic Complex in Patients with Frozen Shoulder

Different types of interventions were performed in various combinations:Warm compresses, TENS, and ultrasounds with scapular PNF (Proprioceptive Neuromuscular Facilitation) interventions [21];Standard treatment with passive mobilization, stretching and physical therapy in combination with scapular mobilization, and end range mobilization [10];Paraffin wax therapy, scapular mobilization, and home exercise program [23].

### 3.5. Professional Figures Involved

Manual therapy interventions in patients with impingement were performed by physiotherapists [1], a physiotherapist with 4 years of experience in manual therapy [19,20] a physiotherapist with 5 years of experience and certification in manual therapy at the Osteopathy School of Madrid DO [22].

The interventions of the therapeutic exercise programs in patients with impingement were performed by a physiotherapist [13,15], by a physiotherapist with 4 years of clinical experience [16], and monitored weekly by a physiotherapist [14].

The interventions of therapeutic exercise programs in patients with frozen shoulder were performed by a physiotherapist with 3 years of experience with performing scapular and end stroke mobilizations [10] and by a physiotherapist with experience in the treatment of frozen shoulder and with a PNF certificate.

#### Type of Control Groups

The included studies established heterogeneous control groups: in the group of patients treated with manual therapy in studies [1,17,19,20], the same thrusts performed in the intervention group were simulated, while in study [22], shoulder strengthening exercises and stretching were combined (Table 1 and Table 2). In those studies including exercise programs for patients with SIS, the control groups performed:Isometric, isotonic in concentric, and eccentric exercise and stretching all focused on the scapula [18];Therapeutic exercises and manual therapy [15];Periscapular strengthening exercises [16];Shoulder stretching and strengthening exercises [14];In one group, false scapular mobilizations were performed, and in another one, strengthening and stretching exercises were performed under supervision [13].

In those studies including exercise programs for patients with frozen shoulder, the control groups performed:Passive mobilization, stretching, and physical therapy [10];Paraffin wax therapy, capsular stretching, and home exercises [23];Physical therapy was used in one group, and physical therapy and strengthening and stretching exercises were used in the other group [21].

### 3.6. Type of Outcome Measures

The outcomes measured in this review were pain, shoulder function, and scapular kinematics. Pain was considered to be the primary outcome in three studies [13,18,23], and function was the primary outcome in another three studies [13,15,16] (Table 1).

The secondary outcomes were pain in two studies [15,16], function in two studies [18,23], and scapular kinematics in three studies [15,16,18] (Table 1).

In other studies, pain, shoulder function, and scapular kinematics were reported as outcome measures, but without specifying whether they were considered to be primary/secondary outcomes (Table 1).

#### 3.6.1. Shoulder Pain

Pain was used as an outcome in all studies except one study [10]. Another study measured pain and function using the same questionnaire [17].

The outcome measures used for each type of intervention are listed below.

For manual therapy interventions in patients with subacromial impingement, these were:Penn Shoulder Score (PSS);Numeric pain rating scale (NPRS);Western Ontario rotator cuff index (WORC);Visual analog scale (VAS).

For therapeutic exercise in patients with subacromial impingement, these were:Verbal numeric rating scale (VNRS);NPRS;VAS.

For therapeutic exercise in patients with frozen shoulder, these were:NPRS;VAS.

#### 3.6.2. Shoulder Function

Function was used as an outcome in all studies. The outcome measures used for each type of intervention are listed below.

Manual therapy interventions in patients with subacromial impingement, these were:Penn Shoulder Score (PSS);Disability of the Arm, Shoulder and Hand (DASH), the normal and Brazilian versions;Global Rating of Change (GROC).

Therapeutic exercise in patients with subacromial impingement, these were:Quick DASH Turkish version and DASH standard versions;Shoulder Pain and Disability index (SPADI-br), Turkish and standard versions;Shoulder disability questionnaire (SDQ).

Therapeutic exercise in patients with frozen shoulder, these were:FLEX-SF;Shoulder Constant Score;Simple Shoulder test.

#### 3.6.3. Scapular Kinematics

Scapular kinematics was used inconsistently in the studies and was not present in two studies [13,23].

The outcome measures used for each type of interventions are listed below.

For manual therapy interventions in patients with subacromial impingement, these were:Digital inclinometer;Flock of Birds hardware;Sistema 3SPACE FASTRAK.

For therapeutic exercise in patients with subacromial impingement, these were:Electromagnetic tracking tools;Digital inclinometer;Infrared cameras.

Therapeutic exercise in patients with frozen shoulder, these were:3SPACE FASTRAK System;Lateral Scapular Slide Test.

### 3.7. Follow-Up

Almost all the studies established short-term follow-ups. The shortest follow-ups were in the post-intervention period [1,17,19,21], while the longest one was after 16 weeks [16]. Specifically, the follow-up timings for each study were: immediately post intervention [1,17,19,21], after 48 h [1,17], after 4 days [20], after 8 days [20], after 10 days [23], after 2 weeks [13], after 3 weeks [13,15], after 4 weeks [10,16,22], after 6 weeks [14], after 7 weeks [13], after 8 weeks [10,16], after 11 weeks [13], after 12 weeks [14], after 3 months [15], and after 16 weeks [16].

### 3.8. Qualitative Analysis on the Effectiveness of Interventions

A qualitative analysis of the results was performed in narrative form, ordered by type of intervention and according to the outcome measure. A tabular representation was made comprising the results reported directly by the patients (function and pain).

### 3.9. Improvements as a Result of Manual Therapy Interventions on Pain, Function, and Scapular Kinematics in Patients with SIS

#### 3.9.1. Efficacy of Manual Therapy in the Improvement of Shoulder Pain in Patients with SIS

Four studies investigated the efficacy of thoracic thrusts on shoulder pain in patients with impingement, and three studies [1,17,19] reported that the pain improved, but there were no significant differences in the control groups who simulated the same type of technique (Table 2).

In one study [20], a significant improvement in pain was found only in the group treated with a true thoracic thrust (Table 2).

Adding manual therapy interventions to shoulder strengthening and stretching exercises compared to a control group that only stretches and strengthens resulted in slightly greater improvements, which can be explained by the fact that the control group started from lower values of pain at rest [22] (Table 2).

#### 3.9.2. Efficacy of Manual Therapy in Improving Shoulder Function in Patients with SIS

All the studies included in the macro-category of “manual therapy” reported similar results regarding the improvement of function, regardless of the type of manual therapy. In studies [1,17,20], in which thoracic thrusts were administered, and in study [22], in which a combination of manual therapy exercises with strengthening and shoulder stretching exercises were performed, there were improvements, which are comparable to those achieved in the various control groups (Table 3).

**Table 3 jfmk-08-00038-t003:** Function scale values in considered studies.

Study	Outcome Measure Tool	Results		
Study [1]	Penn Shoulder Score		Pre-intervention	Post-intervention
		Thoracic thrust (average)	71.8 ± 11.1	80.4 ± 10.9
		Simulated Thrust (average)	70.9 ± 12.5	80.2 ± 11.2
Study [17]	Penn Shoulder Score	Median of changes		
		Supine Thrust	Seated Thrust	Simulated Thrust
		2.0 (1.0–5.0)	2.6 (0.8–6.3)	3.8 (0.0–8.7)
Study [20]	DASH		Differences between initial assessment and 2 day pre-intervention	Differences between initial assessment and follow-up
		Thoracic thrust (average)	−3.9 (−6.3 a −1.6)	−4.6 (−7.2 a −2.0)
		Simulated Thrust (average)	−1.0 (0.8 a −2.9)	−4.7 (−2.1 a −7.4)
Study [22]	DASH		Pre intervento	Post-intervento
		Exercise + manual therapy (average)	25.3 ± 16.1	12.4 ± 12.3
		Only exercise (average)	20.8 ± 10.4	11.7 ± 9.5

#### 3.9.3. Efficacy of Manual Therapy in Improving Scapular Kinematics in Patients with SIS

In studies [1,17,19,20,22], there were no relevant changes in the scapular kinematics.

In studies [19,20], a small improvement in upward scapular rotation capacity was observed in the groups treated with thoracic thrusts compared to that of the control groups. In study [22], in which the intervention consisted of a series of manual therapy techniques in combination with strengthening and stretching exercises of the shoulder, there was an increase in the degree of upward scapular rotation, but there was no significant difference compared to that of the control group.

After thoracic thrust intervention, an increase in the degree of inward scapular rotation was observed in studies [1,19], without differences from that of the control group (Table 3).

### 3.10. Improvements Obtained on Pain, Function, and Scapular Kinematics through Various Therapeutic Exercise Programs, in Patients with SIS

#### 3.10.1. Efficacy of Various Therapeutic Exercises Programs in Patients with SIS in Improving Shoulder Pain

Analysis of the various therapeutic exercise programs that had pain as an outcome revealed that although improvements were observed, they were similar to those of the control groups [13,14,16]. In contrast, adding tactile and verbal feedback to a scapula-focused exercise program was shown to be more effective in improving pain compared to the program of the control group that performed the same exercises without scapular feedback [18]. In the study [15], the group that received scapula-focused treatment had greater pain improvement than the control group did (Table 2).

#### 3.10.2. Effectiveness of Various Therapeutic Exercise Programs in Patients with SIS in Improving Shoulder Function

Analysis of the various therapeutic exercise programs that had shoulder function as an outcome revealed that there were improvements, but they were also similar to those of the control groups [13,14,16] (Table 4).

Instead, adding tactile and verbal feedback to an exercise program focused on the scapula was shown to be more effective in improving function compared to the program of the control group, which performed the same exercises without scapular feedback [18] (Table 4).

Similarly, in study [15], where after nine treatment sessions, the group that received treatment focused on the scapula demonstrated a much greater improvement in function than the control group did, which was treated with therapeutic exercise and manual therapy. In the 3 month follow-up, the function further improved in both groups, but mostly in the control group (Table 4).

**Table 4 jfmk-08-00038-t004:** Function scales values in the considered studies.

Study	Outcome Measure Tools	Results					
Study [13]	DASH		0 week	2 week	3 week	7 week	11 week
		Scapular mobilization group (average)	41.4	32.1	29.7	19.5	28.5
		False scapular mobilizations group (average)	41.7	31.0	25.6	18.8	24.3
		Group with supervised exercises (average)	37.5	29.6	27.2	23.4	20.5
Study [14]	SPADI (disability)		Pre-intervention	6 week		12 week	
		Intervention group (average)	36.08 ± 22.23	16.82 ± 19.59		7.00 ± 10.34	
		Control group (average)	41.58 ± 22.96	24.12 ± 17.26		19.42 ± 20.16	
Study [15]	SDQ		Before treatment			After treatment	
		Intervention group (average)	55.9 ± 14.6			35.0 ± 14.0	
		Control group (average)	50.9 ± 11.9			48.7 ± 11.3	
Study [16]	SPADI-br		Pre-interv	4 week		8 week	16 week
		Intervention group (average)	65.7	43.5		39.7	34.2
		Control group (average)	63.3	49.8		37.1	32.8
Study [18]	DASH		Before treatment			Follow-up	
		Treatment with feedback (average)	26.4			13.2	
		Treatment without feedback (average)	24.4			17.3	
		Control group (average)	23.1			25.0	

#### 3.10.3. Efficacy of Various Therapeutic Exercise Programs in Improving Scapular Kinematics in Patients with SIS

Scapular kinematics following therapeutic exercise programs was an outcome measure that was investigated in four studies [14,15,16,18] (Table 1 and Table 4).

Overall, the improvements were not significant, and they did not occur in all the studies.

In the studies [15,16], there were no changes in scapular kinematics in any group, except for a small change in the upward scapular rotation at 90° of humeral elevation in the follow-up at 16 weeks [16]. This improvement was observed in the humeral stabilization exercise treatment group and not in the control group (Table 1).

In studies [14,18] scapular kinematics changes were more significant:Ref. [18] Greater upward rotation and posterior tilt of the scapula were found in the group in which tactile and verbal scapular feedback was added to the scapula-focused treatment; (Table 1).Ref. [14] Greater external rotation and scapular posterior tilt were observed in the group who performed scapular stabilizations than those in the control group. Furthermore, at follow-ups there were slight improvements in both groups in scapular rotation above 60°–90°–120° of humeral elevation and 90° of humeral lowering. The most significant improvements were noted in scapular rotation above 30° of humeral elevation and 60° of humeral lowering in the intervention group (Table 1).

### 3.11. Improvements Obtained on Pain, Function, and Scapular Kinematics in Various Programs with Therapeutic Exercises in Patients with Frozen Shoulder

#### 3.11.1. Efficacy of Various Therapeutic Exercises Programs in Patients with Frozen Shoulder in Improving Shoulder Pain

Pain was the outcome that was investigated in two studies [21,23], which used therapeutic exercise programs for treatment of frozen shoulder. The various types of interventions were shown to be effective in improving pain:Ref. [23], the pain improved in both groups, but the improvement was significantly higher in the group treated with scapular mobilizations than it was in the control group; (Table 2)Ref. [21], the pain improved in all three groups, but there was a greater improvement in the group treated with scapular PNFs and physical therapy and in the group treated with physical therapy compared with that of the control group treated with classical exercises and physical therapy. An explanation of these results may be due to the fact that the intensity of pain measured before the interventions was greater in the two groups in which greater improvements were found than that in the third group (Table 2).

#### 3.11.2. Efficacy of Various Therapeutic Exercises Programs in Patients with Frozen Shoulder in Improving Shoulder Function

Shoulder function was used as an outcome in three studies [10,21,23] that treated frozen shoulder with various therapeutic exercise programs (Table 5).

In study [21], the improvements in shoulder function were similar in both the intervention and control groups, while in study [23], scapular mobilizations had greater shoulder function improvement than those in the control group did (Table 2).

In the Yang et al. study [10] patients with frozen shoulder were selected based on the criterion that they had at least one of the following three characteristics: less than 97° of humeral elevation, less than 39° of external humeral rotation, and less 8° of posterior scapular tilt. The participants were then divided into three groups: an intervention group that satisfied the criterion, a control group that satisfied the criterion, and a control group that did not meet inclusion criterion. The intervention group that met the criterion was treated with standard manual therapy plus end of range scapular mobilizations, whereas the two control groups received standard manual therapy. At the 8 week follow-up, functional improvement was observed in all groups, including major improvements in the criterion intervention group (Table 5).

**Table 5 jfmk-08-00038-t005:** Function scales values in considered studies.

Study	Outcome Measure Tool	Results			
Study [10]	FLEX-SF		Pre-intervention	4 week	8 week
		Intervention group with criterion (average)	31.7	32.7	39.8
		Control group with criterion (average)	30.8	31.6	32.1
		Control group without criterion (average)	32.8	34.5	37.2
Study [21]	*Simple Shoulder Test*		Pre-intervention	Post-intervention	
		Scapular PNF (average)	6.77	8.16	
		Classic exercises (average)	6.94	8.47	
		Control group (average)	5.94	7.11	
Study [23]	*Constant score*		Pre-intervention	Post-intervention	
		Intervention group (average)	48.69	78.13	
		Control group (average)	48.81	67.16	

#### 3.11.3. Efficacy of Various Therapeutic Exercises Programs in Patients with Frozen Shoulder in Improving Shoulder Kinematics

In study [21], the Lateral Scapular Slide test was used to evaluate scapular dyskinesia. This test is used to evaluate any asymmetry in the distance between the lower corner of the scapula and the closest spinous process in the horizontal plane, which was tested in three different positions.

The results of this test showed no significant differences between the groups (Table 1).

## 4. Discussion

The objective of this systematic review was to compile current knowledge in the scientific literature regarding the efficacy of physiotherapeutic interventions on the scapulo-thoracic complex in the improvement of shoulder pain, function, and scapular kinematics in patients with subacromial impingement and frozen shoulder. The scientific literature regarding the study topic is not limited to the studies included in the review, but it must be considered that the gray literature and non-randomized or non-controlled clinical trials with less evidence were not considered. Furthermore, some studies that may have been relevant to the review were not included due to the inability to obtain the full texts.

Studies that investigated the effectiveness of interventions on specific subpopulations, such as swimmers, volleyball players, and tennis players, were not considered, which further narrowed the research. However, not having predetermined preferential control groups has widened the field of inclusion.

The discussion of the results has been written according to the three outcomes of interest and the type of approach used in the studies in order to discuss any strengths or gaps that exist in the literature.

### 4.1. Pain and Function Outcomes

The thoracic thrusts that were performed in a single session did not prove to be better in terms of pain and function, independent of the thoracic portion treated and of the position of execution, compared to the same interventions performed in a simulated manner [19,20].

There were improvements in pain and function, but they did not depend on the type of treatment; in fact, the placebo was very significant, as demonstrated by similar outcomes obtained in both the treatment and control groups [19,20]. Since subjects were informed of the potential benefits of spinal manipulation, the subsequent expectations of these benefits likely contributed to the placebo effect of analgesia from the treatment [19]. In the study [20], in which two seated thoracic thrust operations were performed, the greatest improvement in pain was shown in the group, in which a true thrust was performed.

There were no significant changes in scapular kinematics in all groups; therefore, it can be reasonably assumed that improvements in pain and function were not supported by a change in scapular kinematics in the articles included in this analysis.

### 4.2. Scapular Kinematics Outcomes

In contrast, the type of treatment used in the study [20] had an impact on the improvement in scapular kinematics. In fact, there was a greater increase in the degree of the upward scapular rotation in the group treated with true thrusts. This introduces avenues for future research, as the effectiveness of this type of intervention would be demonstrated if they were carried out over a longer term. The effectiveness of thoracic thrusts combined with other types of interventions should also be investigated.

In one study, more small improvements were seen in the group which received a combination of more manual therapy treatment and some strengthening and stretching exercises compared with those of the control group, in which the treatment only consisted of strengthening and stretching exercises [22]. Pain and function improved in both intervention groups; however, a slight increase in the degree of scapular anterior tilt was observed in the group treated with manual therapy interventions.

Following analysis of the results, further studies are needed that investigate the effectiveness of a combination of multiple manual therapy techniques and to test their possible benefits on scapular kinematics. Scapular kinematics is an important parameter that should always be considered, as it is assumed that improvement in scapular kinematics will provide symptomatic relief.

Some scapula-focused therapeutic exercise programs [15,18] that involved stretching, mobilization, and active exercises have been shown to be more effective in improving shoulder pain and function compared to the effects felt in the control groups treated with other types of therapeutic exercises [15]. Interestingly, it was seen that adding tactile and verbal feedback to a scapula-focused treatment gives major benefits regarding pain, function, and scapular kinematics [18]; in this trial, two scapula-focused programs were compared, one of which involved scapular feedback. Through the feedback, the scapular movements were better guided, and as a result, the degrees of scapular upward rotation and scapular posterior tilt were increased [18]. These two increased parameters lead us to suppose that they could induce an increase in the amount of subacromial space; thus, resulting in the alleviation of symptoms.

Rehabilitation programs using scapular stabilization exercises combined with periscapular strengthening and stretching [16], shoulder stretching and strengthening [14], or scapular mobilizations [13] have not been shown to be more effective in terms of pain and function than the effects felt by the control groups that performed the same type of exercises without the scapular stabilizations or mobilizations. The treatments analyzed in the studies [14,16] also experienced small and insignificant changes in scapular kinematics. Further investigation should be conducted on the use of these interventions individually to establish their true effectiveness and attempt to limit the placebo effect.

Finally, after a review of the studies included in this macro-category of interventions, providing scapular feedback seems to provide excellent benefits in the three outcomes considered. However, further studies will still be needed to confirm the veracity of the results obtained, as this method was only used in one of the studies that we reviewed.

The use of interventions focused on the scapulothoracic complex, such as scapular PNF techniques and scapular mobilizations [21,23], have been shown to be more effective than the interventions used in the respective control groups are in improving pain and shoulder function.

However, these two studies did not provide results regarding possible changes in scapular kinematics; therefore, it would be useful to perform further studies that try to quantify an objective measure of the true effectiveness of this type of intervention.

Finally, in the study [10], the effectiveness of performing specific end range mobilizations and scapular mobilizations on patients with frozen shoulder who started with poor scapular mobility was confirmed. Performing these interventions on patients that had less than 8° of scapular posterior tilt, less than 97° of humeral elevation, and less than 37° of humeral external rotation during arm elevation increased the extensibility of the shoulder capsule. “Stretching” the compromised soft tissues induced beneficial effects in terms of shoulder function, scapular upward rotation, and scapular posterior tilt [10]. This study emphasizes that targeting a specific subgroup of individuals and choosing a specific treatment based on their specific limitations could yield many benefits and highlights the importance of performing studies aimed at providing targeted evidence. Knowing the best type of treatment for each subpopulation of individuals with a disease could lower the health care costs and reduce the treatment times, allowing members of these subpopulations to live with reduced functional limitations.

Several limitations were encountered while we were compiling this review. Databases or reviews containing gray literature were not considered; moreover, the search for RCTs could be extended on other scientific research databases. Homogeneous control groups were not established, and in this way, the research, although it was broader, cannot be considered as perfectly reliable in terms of the effectiveness of the interventions due to the heterogeneity of the control groups.

## 5. Conclusions

Physiotherapy exercise programs that include scapular motor control training and scapular mobilizations represent valid alternatives in the management of patients with subacromial impingement syndrome. Finally, the positive scapular function and kinematic effects of specific scapular interventions aimed at a particular subgroup of patients with frozen shoulder who were previously identified through a clinical prediction method (Yang et al.) [10] revealed important scenarios to address in future research.

## Figures and Tables

**Figure 1 jfmk-08-00038-f001:**
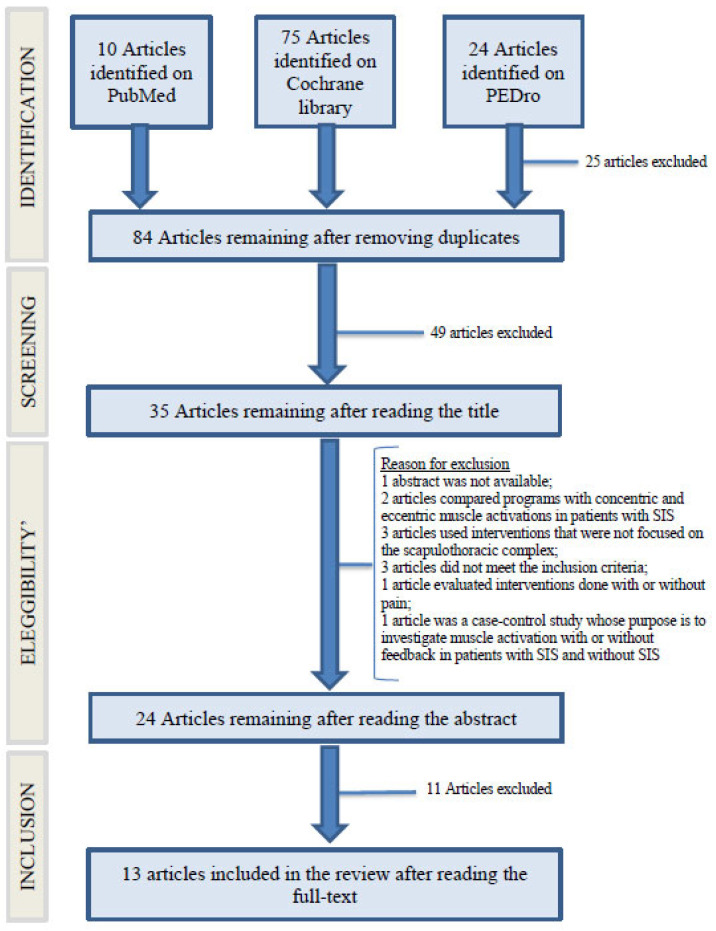
PRISMA (Preferred Reporting Items for Systematic Reviews and Meta-Analysis) flowchart the systematic literature review.

**Table 1 jfmk-08-00038-t001:** Summary table of the studies included in the review.

Author and Year of Publication	Study Design and Level of Evidence (E)	Number (*N*) of Patients and Their Characteristics	Intervention Groups, Control Groups and Number of Treatments	Objectives	Evaluation & Follow-Up	Results
Joseph R. Kardouni et al., 2015 [1]	RCT E = 1b	Patients with impingement; N = 52; Females = 24; Males = 28; Age (years); Experimental group = 18–59 (average 30.8); Control group = 18–59 (average 33.2).	Experimental group n = 26. 3 high-speed, low-amplitude thrusts, one for the low, one for the middle, and one for the high thoracic spine, each performed 2 times. Control group n = 26. Same thoracic thrust, but simulated.	Thoracic and shoulder kinematics during limb elevation: 3SPACE FASTRAK, which is an electromagnetic tracking tool; Pain: *numeric pain-rating scale* (NPRS); Thoracic exursion: electromagnetic capture system; Function: Penn Shoulder Score (PSS).	Immediately before and immediately after treatment, thoracic and scapula kinematics and NPRS were evaluated. From 24 to 48 h later, NPRS, PSS, and *global change rating* (GROC) were assessed.	Thoracic and scapula kinematics: no change in the two groups. Scapulo-thoracic excursion: no change in the two groups. Pain: similar improvements in both groups. Function: Similar improvements in both groups
Jing-Ian Yang et al., 2012 [10]	RCT	Patients with Frozen Shoulder N = 34 (2 did not complete treatment); Males = 12; Females = 22; Age (years); Criterion intervention group= average age 56.8; Criterion control group = average age 54.9; Control group = average age 54.3.	Intervention group with criterion n = 10; passive mobilizations, stretching in flexion and abduction of the shoulder, physical therapy, active exercises, end range mobilizations, and scapula mobilization; Control group of criterion n = 12 passive mobilizations, stretching in flexion and abduction of the shoulder, physical therapy, and active exercises; Control group n = 10 passive mobilizations, stretching in flexion and abduction of the shoulder, physical therapy, and active exercises; All patients were treated twice weekly for 8 weeks.	ROM: measurements made using a standard inclinometer; Disability: completion of a self-administered questionnaire specific to the shoulder, with a fixed index of items, called FLEX-SF. Kinematics of the shoulder: FASTAK motility analysis system.	Evaluations were made before treatment at 4 weeks, and finally, at 8 weeks.	ROM: From 4 and 8 weeks, the humeral external rotation and the ability to put the hand behind the back improved in the control group compared to that of the control group with criterion; at 4 weeks, the ability to put the hand behind the back improved in the intervention group with criterion compared to that of the control group; at 8 weeks, both the ability to put the hand behind the back and the external humeral rotation improved. There were no significant differences between the control group and the criterion and intervention groups. Disability: At 8 weeks, the questionnaire score was better in the control group than that in the criterion control group; at 8 weeks, the questionnaire score was better in the criterion intervention group than that in the criterion control group, while there were no significant differences between the control group and the criterion intervention group. Shoulder kinematics: From 4 and 8 weeks, the upward rotation of the scapula, the posterior tilt and the scapulohumeral rhythm improved in the control group compared to those of the criterion control group; at 8 weeks, the posterior scapula tilt and humeral scapular rhythm improved in the criterion intervention group compared to those of the criterion control group; there were no significant differences between the criterion intervention group and the control group.
Aydan Aytar et al., 2015 [13]	RCT	Patients with SIS N = 66; Males = 15; Females = 51; Age (years); Group treated with scapular mobilizations (GMS) = average age 52; Group treated with false scapular mobilizations (GFMS) = average age 52; Group treated with exercises (GE) = mean age 51.	GMT n = 22 treated with scapula mobilizations; GFMT n = 22 treated with false scapula mobilizations; GE n = 22 treated with muscle strengthening and stretching exercises. In all groups, before performing the listed interventions, patients were received warm compresses and TENS. Treatments were performed for 3 weeks, 3 times a week. After 3 weeks, all groups were instructed to perform GE exercises at home.	*Primary* Function: Quick DASH Questionnaire; Pain: VAS, evaluated at rest, during the night and during activity. *Secondary* ROM: measured the active ROM with a protractor; Satisfaction of the participants: Likert scale.	All assessments were made before the interventions, after two weeks, after three weeks, after seven weeks, and after eleven weeks. The satisfaction questionnaire was administered at the end of the three weeks of treatment.	Pain: there were similar improvements in all three groups. Function: similar improvement to the assessment after the third week, while in subsequent evaluations, it worsened in GMS and GFMS. ROM: similar improvements in all three groups. Participant satisfaction: was greater in GMS and GE.
Elif Turgut et al., 2017 [14]	RCT	Individuals with SIS and scapular dyskinesia participated in the study. N = 30; Males = 16; Females = 14; Age (years); Intervention group (GI): average age 33.4; Control group (GC): average age 39.5	GC n = 15 patients treated with strengthening muscle exercises and stretching; GI n = 15 patients treated with scapular stabilization exercises, followed by shoulder strengthening and stretching exercises. The two intervention programs lasted 12 weeks, monitored weekly by physiotherapist.	Kinematics of the scapula and humerus: measured with an electromagnetic tracking instrument, with measurements made during the elevation and lowering of the arm on the sagittal plane; Function: Turkish version of the Shoulder Pain and Disability Index (SPADI); Pain: VAS, at rest, during the night and during activities.	Assessments were made before intervention, after the sixth week, and immediately after the end of treatments.	Kinematics of the scapula and humerus: increased external scapular rotation during the elevation and lowering of the arm in the GI; increased upward scapular rotation in the GI after 12 weeks; increased rear tilt in the GI; Function: improvements in questionnaire scores at the end of all interventions, but between the two groups there were no significant differences; Pain: decreased pain in the two groups, but without significant differences.
F. Struyf et al., 2012 [15]	RCT	Patients with impingement N = 22; Males = 10; Females = 12; Age (years); Experimental group = 46.2 average age; Control group = 45.4 average age.	Experimental group n = 10 manual mobilizations, stretching, and training on the motor control of the scapula. Control group n = 12. Exercises for strength in eccentric for the muscles of the rotator cuff, manual therapy with multidirectional gleno-humeral mobilizations and ultrasound. Patients in both groups were treated with 9 and 30 min therapy sessions at a frequency of 1–3 per week.	Primary Function: *Shoulder Disability Questionnaire* (SDQ). Secondary Pain: *verbal numeric rating scale* from 0 to 10 (VNRS) for impingement tests and VAS scale for pain at rest and during activity; *Scapula measurements* Visual observation of any winged scapula; Shoulder anteposition: distance between the posterior edge of the acromion and the table in supine patient; Length of the small pectoral muscle; upward rotation of the scapula: inclinometer; motor control of the scapula: kinetic test of internal rotation; isometric force of elevation: dynamometer in the position of the Jobe test.	Assessments were made before treatment, immediately after treatment, and three months after treatment.	Function: significant improvements in the experimental group compared to that of the control group, maintained even in the 3 month follow-up. Pain: reduction of pain in the experimental group at rest, during movement, and also during the Neer test. Scapula position: no change, not even in follow-up.
Gisele Harumi Hotta et al., 2020 [16]	RCT	Patients with subacromial pain N = 60; Males = 18; Females = 42; Age (years); Group with periscapular reinforcement (GRP) = average age 47; Scapular stabilization group (GSS)= average age 51.	GRP n = 30. They were treated with 6 progressive reinforcement exercises for the descending, middle, and ascending trapezius muscles and the anterior serratus muscle; all exercises were performed with progressive loads; GSS n = 30. Same exercises performed by the group as before with in addition six scapular stabilization exercises to emphasize retraction and scapular depression. Treatments in both groups were performed for 8 weeks 3 times a week on non-consecutive days.	*Primary* Function: *shoulder pain and disability index* (SPADI-Br) questionnaire. *Secondary* Pain: *numeric pain rating scale* (NPRS); perception of the effect of treatment: scale of the perceived overall effect; Satisfaction: MedRisk questionnaire; Chisesiophobia: Tampa scale; Force in isometry: hand dynamometer; ROM and scapula position; digital inclinometer.	Assessments were made: before intervention, at 4 weeks, at 8 weeks and 16 weeks.	This study showed that a protocol of progressive periscapular muscle strengthening exercises, with or without scapular stabilization exercises, improved all the parameters that were evaluated, but there were no differences in improvements between the two groups. The only thing that improved the most in the GSS was the upward scapula rotation at 90° of humeral elevation.
Jason K. Grimes et al., 2019 [17]	RCT E = 1	Patients with sub-acromial pain N = 60; Males = 37; Females = 23; Age (years); Group treated with supine thrust (GTSU)= average age 37.6; Group treated with seated thrust (GTSE) = average age 35.6; Group treated with simulated thrust (GTF) = mean age 36.5.	GTSU n = 20 treated with a thrust for the upper portion of the thoracic spine in supine position; GTSE n = 20 treated with a thrust for the upper portion of the thoracic spine while seated; GTF n = 20 treated with a simulated thrust performed while seated. All types of interventions were performed twice.	Pain and function: Penn Shoulder Score (PSS) questionnaire; Evaluation of mobility: inclinometer to measure the various scapular movements both passively and actively, during the elevation of the arm on the scapular plane; the active elevation of the shoulder on the scapular plane was measured using a protractor; Length of the small pectoral muscle: measured with a tape taking reference points; Shoulder strength: measured with a hand dynamometer the strength of the middle trapezius muscle, ascending trapezius muscle, and anterior serratus muscle.	Evaluation of motility, pectoral muscle length, and shoulder strength were evaluated before treatment and immediately after treatment; PSS was administered before treatment and 48 h after treatment.	Pain and function: the PSS improved in all three groups, but between the groups, there were no major differences. Motility: there were no differences between the 3 groups, both for scapular and humeral movements. Length of the pectoral small muscle: there were no significant differences between the 3 groups. Shoulder strength: there were no statistically significant increases in strength after thoracic thrust; no significant differences in strength were observed between the 3 groups.
Mahsa Moslehi et al., 2020 [18]	RCT E = 2	Patients with impingement N = 75; Males = 25; Females = 50; Age (years); Scapula-focused treatment group with feedback (GTSF) = 34.6–42 (38.3 average); Scapula-focused treatment group (GTS) = 33.4–45.5 (37.5 average); Control group = 36.1–40.3 (38.2 average).	GTS n = 25 training on shoulder positioning (first week), strengthening of the rotator cuff muscles (second, third, fifth, sixth, and seventh week), and exercises on flexibility (fourth and eighth weeks); GTSF n = 25 exercises focused on the scapula with tactile and verbal feedback; Control group n = 25 no intervention. The groups were treated for 8 weeks.	Primary Pain: VAS. Secondary Function: questionnaire Disability of the Arm, Shoulder and Hand (DASH); Scapular kinematics: infrared cameras.	Assessments were made before treatment and at the end of 8 weeks.	Pain: in both groups the VAS values improved, but the improvement was more significant in the GTSF. Function: in both groups it has improved, but the improvement is more significant in the GTSF. Scapular kinematics: after 8 weeks, the upward rotation of the scapula and the posterior tilt increased in GTSF, while there were no differences in the control group and in the GTS.
Melina N.Haik et al., 2014 [19]	RCT E = 4	Patients with impingement (PI) and patients without impingement (PSI) N = 97; Age (years); Impingement patients treated with thrust (PIT)= 33.8 average age; Patients with impingement treated with simulated thrusts (PIFT) = 29.7 average age; Patients without impingement treated with a thrust (PSIT) = 25.5 average age; Patients without impingement treated with simulated thrusts (PSIFT) = 26.1 average age.	PI n = 50; PSI n = 47: these two groups of patients were each divided into two groups. PIT n = 25 treated with thrust aimed to the intermediate thoracic tract; PIFT (with SIS) n = 25 treated with a simulated thrust; PSIT (without SIS) n = 24 treated with thrust aimed to the intermediate thoracic tract; PSIFT n = 23 treated with a simulated thrust.	Primary Pain: Western Ontario Rotator cuff index (WORC) and NPRS questionnaire to assess pain during arm elevation and lowering; Function: DASH questionnaire; Scapular kinematics: evaluated using an electromagnetic tracking system during an elevation of the arm on the sagittal plane (Flock of Birds hardware). Secondary Effects on scapular kinematics in groups formed by individuals without impingement.	Assessments were made before intervention and immediately after intervention.	Pain: Improved after thrust in both PIT and PIFT, but the biggest increase was in PIT. Scapular kinematics in the 2 groups of patients with impingement: a few changes were observed, e.g., there was an increase in the degree of the upward rotation of the scapula during arm elevation in the PIT and a slight increase in the degree of internal rotation in the PIT and PIFT. Scapular kinematics in the two groups of patients without impingement: increase in the degree of the upward rotation of the scapula during arm elevation in both groups, more significant in PSIT; a slight increase in the degree of anterior tilt was also observed during arm elevation in the PSIT.
Melina N. Haik et al., 2017 [20]	RCT	The study involved people diagnosed with SIS N = 61; Males = 38; Females = 23; Age (years); Group treated with thoracic manipulation (GMT) = 32.5 average age; Group treated with false thoracic manipulation (GFMT) = 31.3 average age.	GMT n = 30: thoracic thrust to seated patient; GFMT n = 31: similar technique was used, but with thrust applied in different patient position. All operations were performed twice after 3–4 days.	Pain: NPRS and administration of the WORC questionnaire; Function: administration of the DASH questionnaire; Scapular kinematics: measured by a computerized system during the complete elevation and lowering of the limb on the sagittal plane. Muscle activation: measured the activity of the ascending, middle and descending trapezius muscle and the anterior serratus muscle with electromyography.	Evaluations were made before the first intervention, before the second intervention, immediately after the second intervention, and then, 3–4 days after the second intervention.	Pain: a decrease in pain was observed in GMT before the second intervention and immediately after the second intervention; there were minimal improvements in the WORC and no difference between the two groups. Function: minimal improvements and no difference between the two groups; Scapular kinematics: in the GMT after the two interventions, there was an increase in the degrees of upward rotation and scapular tilt, while in the GFMT, the anterior scapular tilt increased after the second intervention; 3–4 days after the interventions, the difference between the two groups in the upward scapular rotation was maintained, even if it was minimal. Muscle activation: activation of the descending trapezius after intervention increased in GFMT, while activity of the middle and ascending trapezius and anterior serratus decreased in both groups.
Nilay Comuk Balci et al., 2016 [21]	RCT	The study involved patients with unilateral frozen shoulder N = 53; Males = 13; Females = 40; Age (years); PNF treatment group (GPNF) = 56.7 average age; Group treated with classical exercises (GEC) = average age 58.1; Control group (GC) = 58.6 average age	GC n = 17 patients were treated with warm compresses, TENS and ultrasound; GEC n = 18 group treated with the same interventions done in the GC plus stretching exercises such as Codman’s movements, and reinforcement exercises that include scapular elevation and scapular stabilization exercises; GPNF n = 18 patients were treated in the same way as GC, with additional scapular PNF techniques. The intervention lasted an hour.	Pain: VAS; Scapular dyskinesia: lateral scapular slide test; Active ROM: protractor to measure abduction and flexion in supine position; Function: Simple Shoulder test (SST), which is a questionnaire.	Assessments were made before the various interventions and immediately after the end of each treatment.	Pain: pain improved the most in GC and GPNF; Scapular dyskinesia: no improvements were found; Active ROM and Function: there were improvements in all three groups, without there being any significant differences between the groups.
Paula R. Camargo et al., 2015 [22]	RCT E = 1b	Patients with impingement N = 46; Males = 24; Females = 22; Age (years); Manual therapy intervention group (GITM): 35.96 average age; Intervention group (GI): 32.65 average age.	GI n = 233 strength exercises and 3 stretching exercises; GITM n = 23 same exercises performed by the GI with an additional 45 min of manual therapy. All patients were treated for 4 weeks.	Scapular kinematics: measurement made through a machine (Flock of Birds hardware) during the elevation of the arm on the scapular and sagittal plane; Function: Brazilian version of the DASH questionnaire; Pain: VAS scale; Mechanical sensitivity: threshold pressure point (point where pressure turns into pain).	The evaluations were made one week before the intervention, and at the end of the 4 weeks of intervention.	Scapular kinematics: there were no clinically relevant changes in any group. A noticeable increase in the degree of anterior tilt was observed after GITM intervention. Function: both groups had improvements in the questionnaire score, which were more relevant in the GITM. Pain: after the interventions, it improved in both groups, although the most significant improvement occurred in the GITM. Mechanical sensitivity: mechanical sensitivity in the shoulder region is improved in both groups after interventions.
S. Arul Pragassame et al., 2019 [23]	RCT	The study involved patients with unilateral frozen shoulder N = 30; Males = 19; Females = 11; Age (years); Group treated with scapula mobilizations (GMS): average age 51.73; Group treated with conventional therapy (GTC) = average age 51.40	GTC n = 15: this group was treated with paraffin wax before each surgery, capsular stretching, and home exercises; GMS n = 15: this group was treated the same as the GTC group with the addition of scapular mobilizations. The total duration of both treatments was 10 days, one session per day.	*Primary* Pain: NPRS ROM: measured the active abduction and external rotation on the scapula plane with a protractor; *Secondary* Function: shoulder constant score, (a questionnaire to evaluate shoulder function).	Assessments were made before treatment and immediately after the 10 days of treatment.	Pain: pain improved significantly in both groups, but the improvement was most significant in GMS; ROM: both abduction and external roation improved in both groups, but more so in GMS; Function: improved in both groups but more in GMS.

## Data Availability

Data available on request due to restrictions (privacy and ethical).

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
