# Peer review of "Effectiveness of Therapeutical Interventions on the Scapulothoracic Complex in the Management of Patients with Subacromial Impingement and Frozen Shoulder: A Systematic Review"

_jfmk, 2023, doi:10.3390/jfmk8020038_

Round 1

Reviewer 1 Report

This systematic review summarized the evidence on the effects of physiotherapy techniques in the treatment of shoulder pain, particularly shoulder impingement syndrome and frozen shoulder. The authors found that exercises, including motor control exercises and mobilizations have efficacy in managing symptoms due to above mentioned conditions. The manuscript is comprehensively presented. However there are issues that should be addressed to improve the readability and presentation.

1- Abstract, conclusion should also consider the role of scapulo-thoracic complex and scapula kinematic function as stated in the Introduction, last para.

2- Results should be summarized and succinctly presented with a focus on important and relevant findings. Repetitions with Tables should be avoided. Tables should be kept to important ones. Some Tables could be considered as appendix.

3- Page 5, Table 1, the level of evidence should be defined in the Method section.

4- Page 13, Enrollment, line 199, I did not find it in the Table 1.

5- Page 13, lines 223 and 225, what do you mean "13capula-thoracic"?

6- Page 17, Table 2, NPRS is an ordinal measure and should be reported as "Median (IQR). Table 2, Table 4, and Table 6 seem similar; please clarify it.

7- Discussion, please subheadings based on the review purposes.

Author Response

Thank you for your important suggestions. I underlined in the text the changes made.

  1. In the abstract, I added the required information.

  1. I summarized the results where possible and joined tables 2-4 and 6 that became a single table (table 2), while table 5 became table 4 and table 7 became table 5.

  1. The levels of evidence have been defined in the methods section as required.

  1. Typing error.

  1. Typing error.

  1. I have modified Table 2, each Table refers to several Studies: Table 2 for studies that have studied the effectiveness of thoracic thrusts, Table 4 for studies that have studied the effectiveness of various programs of therapeutic exercises in patients with SIS, Table 6 for studies that treated patients with scapular PNFs.

  1. I have subheadings based on the review purposes as required.

We hope I have fulfilled your requests and we hope to work with you in the future.

Reviewer 2 Report

Ferlito et al. submitted a systematic review about effectiveness of the following therapeutical interventions : (1) manual therapy in subacromial impingement syndrome (2) exercise programs in subacromial impingement syndrome  and  (3) exercise programs in patients with frozen shoulder.  In the introduction they describe pathomechanisms of these conditions and the impact of these conditions on patient's daily life.  They evaluated the following outcome measures: pain, shoulder function and scapular kinematics.

The study was conducted based on PRISMA statement guidelines  for systematic reviews. Completeness of the topic covered seems adequate: records were identified through search in PUBMED, PEDro and Cochrane library, considering only randomized clinical trials in the period of 2011-2021. Inclusion of only randomized clinical trials which provides high quality of evidence  and consequently augments the quality of this review.   Flow chart  diagram of the studies selection is provided  and eligibility criteria are clearly defined.   

The topic covered is relevant for the scientific community and clinical practice.

The references are appropriate. Conclusions are supported by listed citations.

The review is comprehensive and of relevance for the field.

Author Response

Thank you for your evaluable comments and appreciations.

Round 2

Reviewer 1 Report

MAJOR REVISION IS NEEDED.

1- Abstract, Conclusion should consider the frozen shoulder. The role of the scapulo-thoracic complex should be evaluated. I did not see findings on this matter. This should be noted in the review.

2- When defining a term to use as an abbreviation, please apply it throughout the manuscript (e.g., Introduction, last line, line 86,  SIS should be used).

3- Page 3, 2.2. Studies included, this section could be omitted for repetition with Table 1).

4- Table 1, page 4, criteria for evidence-based design quality should be defined.

5- Page 11, Results, "significant."

6- Page 12, lines 172-175, it is an incomplete sentence (Although ...who received.)

7- Page 12, Sample and Gender, the results on these could be moved to Table 1. Section Age could be omitted.

8- Page 13, lines 215-216; please provide a reference.

9- Page 13, Approaches could be added in Table 2.

10- Page 13, line 230, SIS has been defined before in the Introduction; please use the abbreviation "SIS" here.

11- Page 15, lines 316-317, line 322, line 326, etc., why capital letters?

12- Table 2, Median (IQR) should be reported for an ordinal measure on NPRS.

13- Page 17, Table 3, the title is similar to that of Table 4; please revise it specifically to indicate what the studies are about.

14- Page 20, Table 5, the title should be amended to include frozen shoulder to clarify it specifically.

15- Page 21, Discussion, line 532, please report the effect sizes for the interventions used in the studies included in the review.

16- Page 23, Conclusion, line 603, "through"? 

Author Response

Thank you for your important suggestions, you will find in the text the required changes.

Best,

Vito Pavone

Round 3

Reviewer 1 Report

I would like to thank the authors for revising the manuscript, which now looks much better. There are still minor issues, as outlined below:

1- In table 1, the level of evidence should be defined; it is not addressed despite commenting on it from the previous round of reviews.

2- Table 2, NPRS scores should be reported as Median (IQR).

3- TableS 3 and 4, titles, please define SIS. 

Author Response

Dear Reviewer, thank you again for your evaluable comments. 

Here you will find a point by point responses

Q1- In table 1, the level of evidence should be defined; it is not addressed despite commenting on it from the previous round of reviews.

A1 - In table 1, we added a column with level of evidence for each study

Q2- Table 2, NPRS scores should be reported as Median (IQR).

A2 - Medians of NPRS scores have been added in Table 2

Q3- TableS 3 and 4, titles, please define SIS. 

A3 - SIS has been defined in tables 3 and 4.

We hope you will find the manuscript suitable for publication.